# An Expanding Toolkit for Heterochromatin Repair Studies

**DOI:** 10.3390/genes13030529

**Published:** 2022-03-17

**Authors:** Chetan C. Rawal, Nadejda L. Butova, Anik Mitra, Irene Chiolo

**Affiliations:** Molecular and Computational Biology Department, University of Southern California, Los Angeles, CA 90089, USA; crawal@usc.edu (C.C.R.); butova@usc.edu (N.L.B.); anikmitr@usc.edu (A.M.)

**Keywords:** pericentromeric heterochromatin, homologous recombination, nuclear dynamics, double-strand break repair, repeated sequences, genome stability

## Abstract

Pericentromeric heterochromatin is mostly composed of repetitive DNA sequences prone to aberrant recombination. Cells have developed highly specialized mechanisms to enable ‘safe’ homologous recombination (HR) repair while preventing aberrant recombination in this domain. Understanding heterochromatin repair responses is essential to understanding the critical mechanisms responsible for genome integrity and tumor suppression. Here, we review the tools, approaches, and methods currently available to investigate double-strand break (DSB) repair in pericentromeric regions, and also suggest how technologies recently developed for euchromatin repair studies can be adapted to characterize responses in heterochromatin. With this ever-growing toolkit, we are witnessing exciting progress in our understanding of how the ‘dark matter’ of the genome is repaired, greatly improving our understanding of genome stability mechanisms.

## 1. Introduction

Pericentromeric heterochromatin (hereafter ‘heterochromatin’) occupies about 30% of fly and human genomes [1,2,3,4] (Figure 1), is enriched for silencing epigenetic marks (i.e., H3K9me2/3, and *Drosophila* HP1a [5] or mammalian HP1α/β, and is mostly composed of repeated DNA sequences. For example, in *Drosophila*, about half of these sequences are ‘satellite’ repeats (mostly 5-base-pair sequences repeated for hundreds of kilobases to megabases) and the rest are transposable elements, scrambled repeats, and only about 250 isolated genes [2,3,4] (reviewed in [6,7]).

Pericentromeric repeats are functionally and structurally distinct from lamina-associated domains (LADs) [8], which are distributed along the chromosome arms (Figure 1). In contrast to LADs, pericentromeric sequences are not usually associated with the nuclear periphery (see, for example, refs. [7,9,10,11,12,13,14,15]). Pericentromeric repeats are also distinct from Polycomb-mediated silenced regions, which are typically enriched for H3K27me3 and encompass LADs and other silenced domains [16] (reviewed in [17,18]). Pericentromeric heterochromatin is absent in budding yeast; thus, repair in these sequences has mostly been studied in *Drosophila*, mouse, and human cells.

**Figure 1 genes-13-00529-f001:**
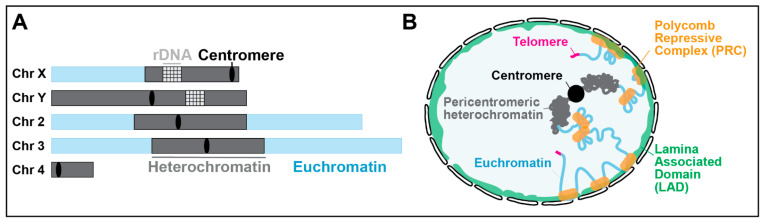
Chromosomal and nuclear position of pericentromeric heterochromatin. (**A**) Schematic view of *Drosophila* chromosomes showing the position and extent of pericentromeric heterochromatin (adapted from [5]). The Y chromosome is mostly heterochromatin. (**B**) Position of pericentromeric heterochromatin relative to the nuclear periphery in a *Drosophila* nucleus, which is distinct from LADs, telomeric repeats, and Polycomb-repressed domains (adapted from [7,18]).

Given the abundance of repeated sequences, heterochromatin poses unique challenges to homologous recombination (HR) repair of DNA double-strand breaks (DSBs) (reviewed in [6,7,19]). DSBs are mostly repaired by HR or non-homologous end joining (NHEJ) (Figure 2). NHEJ is characterized by the direct re-joining of the two ends, which frequently generates small mutations at the repair site [20]. Conversely, HR initiates with a resection to form 3′ single-stranded DNA (ssDNA) that invades ‘donor’ homologous sequences used as templates for DNA synthesis and repair [21]. In single-copy sequences, a unique donor is present on the sister chromatid or the homologous chromosome, and HR is largely ‘error free’. In heterochromatin, however, the availability of up to millions of potential donor sequences associated with different chromosomes can initiate unequal sister chromatid exchange, or intra/inter-chromosomal recombination, leading to deletions, duplications, translocations, release of extrachromosomal DNA circles, or the formation of dicentric or acentric chromosomes [10,15,22,23,24,25,26] (reviewed in [6,7,19,27]).

Despite the risk of aberrant recombination, HR appears to be a preferred pathway for heterochromatin repair in S/G2, i.e., when both HR and NHEJ are available [10,12,28]. Studies across different model systems have identified specialized mechanisms enabling ‘safe’ HR repair in heterochromatin, while preventing aberrant recombination. In *Drosophila* cells, where heterochromatin forms a distinct nuclear domain [10,14], HR repair of heterochromatin requires the relocalization of repair sites to the nuclear periphery before recruitment of the strand invasion component Rad51 [10,15,24,25,26] (reviewed in [6,29,30,31]). Similarly, in mouse cells, DSBs relocalize to the periphery of heterochromatin domains (called ‘chromocenters’), before Rad51 recruitment [11,12,25,32] (reviewed in [6,7]). In both contexts, relocalization likely promotes ‘safe’ HR repair while preventing aberrant recombination by moving the broken site and its repair templates (on the homologous chromosome or the sister chromatid [25,33]) away from the bulk of other repeated sequences before strand invasion (reviewed in [6]).

Various techniques have been developed to study DSB repair, as recently reviewed [34]. Specific approaches have also been applied to the study of DSB repair in other ‘silent’ genomic regions, such as telomeres [35], LADs [36], Polycomb-rich regions [37,38], other silenced repeats [39], and untranscribed loci (inactive genes or intergenic regions) [40] (reviewed in [18]). Additional studies have characterized the responses in pericentromeric regions to damage other than DSBs, such as from UV exposure [41], and heterochromatin replication or over-replication [42,43], and will not be covered by this review. Here, we focus on the main approaches that have been applied to the characterization of DSB repair in pericentromeric regions (Figure 3), including the main conclusions derived from different studies, and the strengths (+) or limitations (−) of different techniques. We also propose how other methods developed to study repair in euchromatin can be adapted to study heterochromatin repair.

## 2. Approaches to Study Heterochromatic DSB Responses

### 2.1. Imaging of Ionizing Radiation (IR)-Induced Repair Foci

HR repair can be followed cytologically because the recruitment of many HR repair components to DSBs gives rise to visible foci [10,44,45]. The formation and disappearance of different repair foci reflects HR progression, including: DSB detection and signaling (e.g., Mdc1 or *γ*H2AX foci in mammalian cells, and Mu2 or *γ*H2Av foci in *Drosophila* cells [10,15,25]); resection (e.g., RPA, ATRIP, TopBP1 [10]); Rad51 loading; and strand invasion (e.g., Brca2, Rad51, or Rad54 foci [10]) (see Glossary and Figure 2). Heterochromatin also forms distinct domains in *Drosophila* and mouse cells, which can be visualized by tagging or staining for heterochromatin-specific proteins or chromatin marks (e.g., HP1 proteins or H3K9me2/3 [9,10,15,25,26,46], Figure 3), enabling characterization of repair kinetics and dynamics relative to heterochromatin domains of these model systems.

Live and fixed cell imaging of IR-induced repair foci has enabled major advances in our understanding of the molecular mechanisms involved in heterochromatin repair [9,10,11,15,24,25,26]. For example, live imaging of *Drosophila* cells revealed that Mdc1/Mu2 and ATRIP or TopBP1 foci form with high efficiency inside the heterochromatin domain [10,15,25], reversing the initial assumption that DNA damage detection, signaling, and resection are delayed by compaction and silencing inside the domain [47,48] (Figure 4). ATRIP and TopBP1 foci appear sooner and are brighter in heterochromatin relative to euchromatin, suggesting that the HR steps leading to resection occur even faster in heterochromatin [10]. These studies also revealed a striking relocalization pathway, with repair foci leaving the heterochromatin domain and reaching the nuclear periphery 1 h after IR, where Rad51 is recruited [10,15,24,25] (Figure 4).

A combination of live imaging and RNAi depletion studies in *Drosophila* cells established that SUMOylation by the heterochromatin-enriched complex Smc5/6, its SUMO-E3 ligases (Nse2/Cerv, and Nse2/Qjt), and the E3 SUMO ligase dPIAS, is required to halt HR progression inside the heterochromatin domain [10,15,24]. At the nuclear periphery, repair sites associate with nuclear pores and inner nuclear membrane proteins (e.g., the LINC complex subunits Koi and Spag4), where a SUMO-targeted ubiquitin ligase (STUbL)/RENi complex likely mediates the repair restart through the ubiquitination of SUMOylated proteins (whose identity is still unknown) (Figure 4), followed by their degradation or reactivation [15].

**Figure 3 genes-13-00529-f003:**
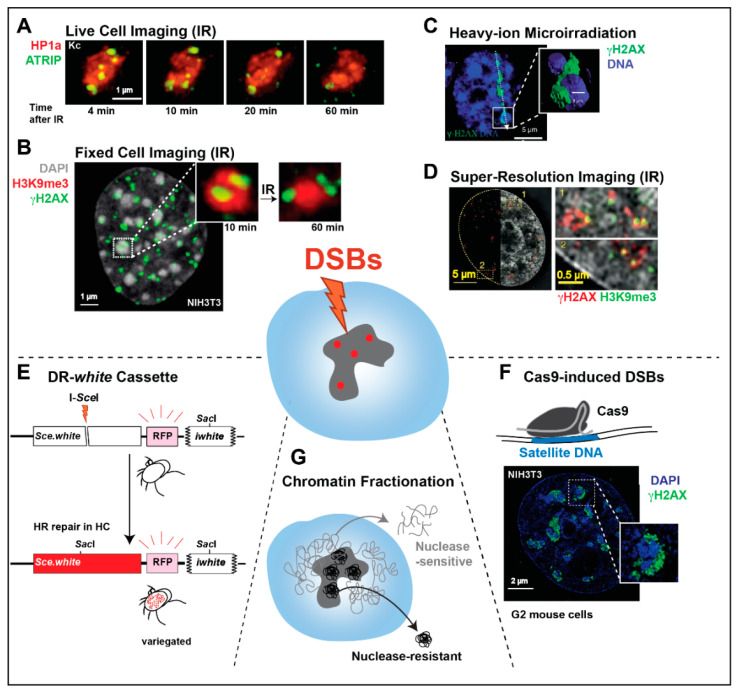
Summary of the main approaches used to study heterochromatic DSB repair. (**A**) Live imaging of *Drosophila* Kc cells treated with a dose of 5 Gy X-rays shows the relocalization of ATRIP foci to outside the heterochromatic domain (marked by HP1a), at the indicated time points after ionizing radiation (IR) (image from [10]). (**B**) IF analysis of mouse NIH3T3 cells fixed 10 min after exposure to a dose of 5 Gy X-rays shows *γ*H2AX foci inside DAPI-bright chromocenters. The magnified detail also shows foci predominantly outside the domain 60 min after IR. (**C**) Image and 3D reconstruction of a mouse embryonic fibroblast (MEF) cell 60 min after exposure to heavy-ion irradiation shows a linear damage streak (*γ*H2AX) bent around a DAPI-bright chromocenter (image adapted from [32]). (**D**) IF analysis by 3D-SIM of *γ*H2AX and H3K9me3 signals 24 h after exposure to a dose of 10 Gy X-rays shows damage foci associated with silenced sequences in HeLa cells (image from [49]). (**E**) HR repair of a DR-*white* cassette inserted in pericentromeric sequences of *Drosophila* results in variegated-red eyes in the progeny. (**F**) IF analysis of Cas9-induced DSBs in the major satellite DNA of mouse NIH3T3 cells shows *γ*H2AX signals at the periphery of DAPI-bright chromocenters during HR repair (image from [12]). (**G**) Chromatin fractionation techniques can enrich nuclear extracts for MNase-resistant heterochromatin, enabling the characterization of heterochromatin-associated proteins during repair.

In addition, the histone demethylase Kdm4A is required to lower H3K56me3 levels in response to IR in *Drosophila* cells, and for the relocalization of heterochromatic repair foci, suggesting a role for Kdm4-mediated H3K56me3 reduction in these dynamics [50].

Live and fixed cell imaging studies also identified a striking network of transient nuclear actin filaments (F-actin) responsible for the directed motion of repair sites to the nuclear periphery [6,25,29,51,52] (Figure 4). Filaments are assembled at heterochromatic DSBs by the actin nucleator Arp2/3 and extend toward the nuclear periphery [25]. Relocalization is also mediated by nuclear myosins (i.e., Myo1A, Myo1B, and MyoV) that interact with Smc5/6, and are activated by Unc45 downstream from Smc5/6, to promote the directed motion of repair sites along the filaments [25]. Critically, extensive relocalization and directed motions are not typically detected at euchromatic DSBs [25]. Myosins are also not involved in the motion of euchromatic repair foci, revealing that long-range motions are specifically important for the repair of heterochromatin, and not of euchromatin [25].

**Figure 4 genes-13-00529-f004:**
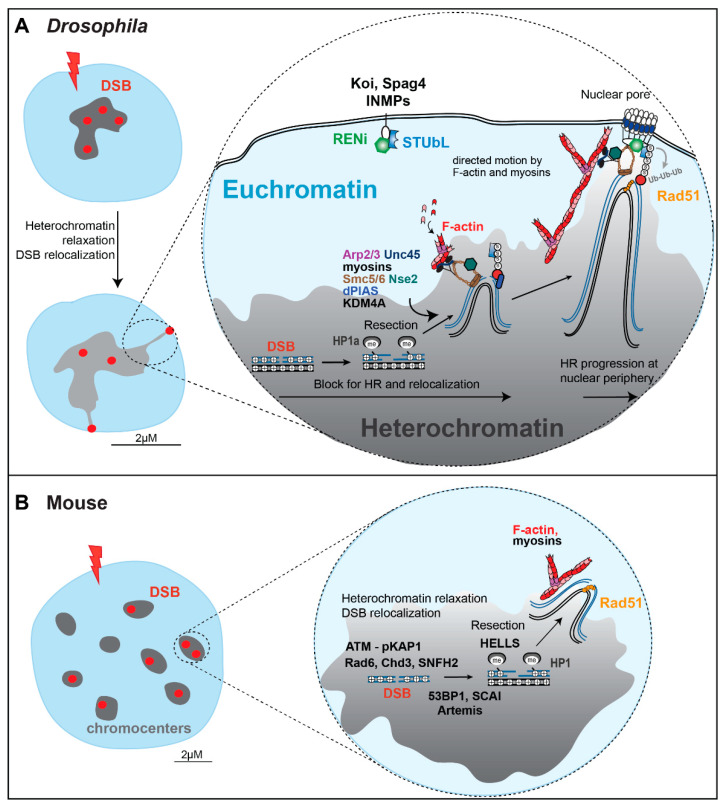
Heterochromatin repair mechanisms in Drosophila and mouse cells. (**A**,**B**) Molecular mechanisms mediating HR repair in *Drosophila* and mouse cells, with some of the main factors regulating DSB resection, chromatin remodeling, heterochromatin relaxation, relocalization of repair sites, and HR progression (also reviewed in [6]). S: SUMOylation. Ub: Ubiquitinaltion. INMP: Inner nuclear membrane proteins. See details in the text.

Importantly, for these studies live imaging needs to be tightly calibrated to enable long kinetics with sufficient time points for mean-square displacement (MSD) analyses. Successful execution of these experiments relies on careful selection of fluorescent tags and microscopy techniques that are particularly gentle on the samples, such as working with a bright-field fluorescent microscope and applying deconvolution processing to the images [25,51,52].

These studies and computer simulations also established that traditional MSD analyses of repair focus tracks are unable to detect directed motions in the context of mixed trajectories, in which directed motions start asynchronously for a population of foci, and in which directed motions are preceded and followed by sub-diffusive motion [6,25,30,52]. Instead, dedicated methods that identify the time intervals of directed motions within longer trajectories need to be applied to these contexts [25,30,52].

Fixed cell imaging studies in mouse NIH3T3 cells show that relocalization of IR-induced heterochromatic repair foci relies on actin polymerization and myosins also in this context [25], revealing conserved pathways between *Drosophila* and mammalian cells [25].

In addition, live imaging of HP1 proteins in response to IR identified a significant expansion of heterochromatin domains in *Drosophila* and mouse cells, corresponding to a volume increase of up to 50% in domain size [10,26,49]. This is dependent on checkpoint (particularly ATR) and resection in *Drosophila* cells [10]. While the significance of this expansion remains unclear, it might reflect a relaxation of the domain to promote damage signaling [46], repair protein accessibility, or increased dynamics [6,10].

Finally, fixed cell imaging analysis of *γ*H2AX foci in mouse cells shows that heterochromatic DSBs are repaired with slower kinetics than euchromatic DSBs; thus, foci remaining in the nuclei 8 h after IR are mostly associated with chromocenters [53,54]. Repair kinetics are further delayed by ATM loss specifically in heterochromatin, revealing the importance of ATM in heterochromatin repair [9]. Cell imaging and genetic approaches have identified Kap1 as a critical phosphorylation target in this pathway, leading to Kap1 release from the chromatin, and chromatin relaxation [9]. Similar approaches have also identified the importance of SCAI in ATM signaling for heterochromatin repair [55].

Building on the discovery that persistent DSBs in response to IR are mostly heterochromatic, and that their resolution is facilitated in the absence of Kap1, several labs have identified additional heterochromatin repair components in human cells, using genetic approaches. These studies support a role for Cdh3 dispersal and Acf1-Snf2h activation in heterochromatin relaxation during repair [53,54], the importance of Artemis and 53BP1 in promoting HR in heterochromatin in G2 [28,56], and a role for HELLS in DSB resection in heterochromatin [57].

(+) A major advantage of using IR is that this treatment induces damage in a nearly synchronous manner, facilitating the detection of relocalization events that occur in relatively short time intervals (e.g., within 1 h from damage induction). IR also induces DSBs with high efficiency in both euchromatin and heterochromatin, and dose titration enables DSB induction at physiological levels. Using *Drosophila* or mouse cells provides unique advantages for imaging approaches, given the organization of heterochromatin in distinct nuclear domains that are easily detectable cytologically. Live cell imaging in response to IR enables the study of 3D dynamics, identifying relocalization pathways and directed motions. Responses to breaks in euchromatin and heterochromatin can be studied within the same cells, enabling direct comparisons and the detection of heterochromatin-specific responses. Another advantage is that the experiments can be performed from very few up to large numbers of cells, facilitating both genetic and biochemical approaches. For fixed and live cell imaging studies, immunofluorescence (IF) or time-lapse approaches are typically completed within a day, providing a quick readout. IR has also been coupled with fluorescent in-situ hybridization (FISH) to follow individual satellites [10]. In *Drosophila*, highly efficient and cost-effective RNAi techniques further facilitate a systematic exploration of the pathways involved in the relocalization of heterochromatic DSBs. *Drosophila* Kc cells are also mostly in S/G2, facilitating the study of HR repair. Large cell size facilitates studies in mouse cells, although individual chromocenters are smaller than the *Drosophila* heterochromatin domain.

(−) IR exposure induces a number of other damages in addition to DSBs [58]. Some repair proteins do not form foci, or form foci that are not visible in live imaging (including most NHEJ components), thus they cannot be studied with these approaches. The techniques described here are mostly applicable to cells with clearly defined ‘domains’ (e.g., mouse and *Drosophila* cells) [52], limiting what can be studied in human cells. Mouse fibroblasts used in the studies mentioned above are also particularly flat, resulting in low resolution along the Z stacks and more limited information on focus dynamics than what can be defined in *Drosophila* cells. It is also critical to choose the right imaging conditions to maximize image collection for focus tracking while minimizing photodamage and photobleaching [51,52]. Live imaging is particularly difficult with multiple channels, resulting in the use of fluorescent tags that are typically less bright and less photostable than GFP (e.g., YFP or BFP variants). This limits the number of components that can be simultaneously imaged in each experiment, reducing the amount of information that can be extracted. Live imaging is also more difficult in tissues [26], where more photo-damaging imaging techniques need to be applied. Further, IR exposure requires specific equipment (i.e., X-ray machine or ^137^*Cs*-irradiator), which might not be available to many laboratories. Once data is collected, quantification using manual or even semi-automated methods is time consuming [51,52]. Finally, random DSB induction does not enable the characterization of chromatin and DNA damage responses with chromatin immunoprecipitation (ChIP)-type approaches, limiting the resolution of these studies.

### 2.2. Imaging of Spatially Defined DSBs Induced by Laser or Heavy-Ion Irradiation

Live and fixed cell imaging have also been applied to DSBs induced along linear paths by heavy-ion irradiation or laser beams crossing heterochromatin domains, where knowing the spatial positioning of the damage facilitates the characterization of kinetic and dynamics of the damage response. For example, when heavy ions traverse mouse chromocenters or human perinucleolar heterochromatin, linear *γ*H2AX streaks bend around the heterochromatin domains, consistent with damage relocalization [32] (Figure 3). TUNEL, *γ*H2AX, and RPA IF signals are visible inside the domain shortly after irradiation, indicating that early DSB formation, signaling, and resection, occur quickly inside the chromocenters [32], similar to what is observed in *Drosophila* cells after IR treatment. This study also confirmed the importance of ATM in resolving heterochromatic DSBs in mouse cells, as ATM^−/−^ MEFs display persistent DSBs associated with the chromocenters. However, similar to what is observed in *Drosophila* cells, ATM is not required for relocalization [32]. Additionally, the progressive reduction in DNA signals at the sites hit by ion radiation suggests local de-condensation of the chromatin at repair sites [32].

Laser beams present an alternative method to induce spatially-defined DSBs in heterochromatin. DSBs are typically induced using a UV laser mounted on the microscope, after cell sensitization with a DNA intercalator (e.g., Hoechst) or a nucleotide analog (e.g., BrdU) [59]. Alternatively, damage can be directly induced without cell sensitization using a near-infrared (NIR) pulsed laser [60]. Live imaging of mouse cells, in response to UV laser-targeting of the chromocenters (marked by HP1β-GFP), revealed heterochromatin expansion and the importance of HP1β phosphorylation by casein kinase 2 (CK2) in this response [46]. It also established the function of HP1β phosphorylation in *γ*H2AX induction, consistent with a role for heterochromatin relaxation in DSB signaling [46]. Chromocenters have also been imaged with energy-filtering transmission electron microscopy after laser-induced damage in mouse cells, which detected heterochromatin relaxation as a 30–40% reduction in the density of chromatin fibers proximal to DSBs [61].

In *Drosophila* cells, DSBs induced inside heterochromatin domains by NIR laser treatment result in *γ*H2Av foci inside the domain shortly after damage [10]. At later time points, sites enriched for Rad51 are mutually exclusive with HP1a signals, suggesting local HP1a release or heterochromatin de-condensation during HR progression [10]. However, surprisingly, laser-induced DSBs do not relocalize to outside heterochromatin domains in this context, suggesting that either breaks induced by laser radiation do not relocalize efficiently, or the high energy associated with laser radiation disrupts some aspect of the relocalization pathway [10].

(+) A clear advantage of these approaches is that the damage is directed across the domain or outside the domain in a precise manner, facilitating the identification of heterochromatin-specific responses. Comparisons between responses in euchromatin and heterochromatin can be conducted within the same cells, reducing confounding effects from cell-to-cell variability, and highlighting heterochromatin specific responses. Relocalization of DNA breaks in response to heavy-ion irradiation is easy to quantify as the beam ‘bends’ around the chromocenter. Generating a high number of DSBs within a short distance also amplifies damage signals, thus facilitating detection of low-abundance proteins by IF or live imaging [59]. Microscope-mounted laser radiation enables fast post-damage imaging and detection of very early repair steps or transient protein recruitment. Finally, as mentioned in the previous section, detection techniques with live or fixed cell imaging are fast, providing a quick readout of experiments.

(−) The high energy associated with heavy ions or lasers can affect other components, including proteins responsible for relocalization [10,62]. Similar to IR, heavy ions and lasers induce multiple types of DNA lesions in addition to DSBs, complicating the analysis of DSB-specific responses [59,63]. Additionally, both lasers and ion irradiation induce multiple DSBs in close proximity [59,63,64,65] (clustered DSBs), which behave differently from sparsely-induced DSBs, potentially inducing non-physiological responses [59,64,66]. For example, the complex damage and clustering of multiple breaks induced by high-LET results in lower usage of NHEJ and higher reliance on HR (reviewed in [66]). Further, the sensitization of cells with a thymidine analog or a DNA intercalator for UV laser treatment [59,64] can potentially affect the DNA and chromatin structure, altering DNA damage responses [67,68]. In these techniques, damage is induced without sequence specificity and at low resolution, preventing the application of ChIP-like approaches to the characterization of damage-induced changes in chromatin responses and protein recruitment. Laser or heavy-ion irradiation are also difficult to apply to small nuclei (i.e., *Drosophila*), given the relatively large size of the beam. Finally, specialized equipment is required for heavy-ion radiation (synchrotron), or for applying a laser mounted on a microscope, which might not be readily available in many laboratories.

### 2.3. Super-Resolution Imaging of Repair Responses in Human Heterochromatin

Heterochromatin repair is particularly difficult to study with cytological approaches in human cells, where pericentromeric regions are not clustered into cytologically distinct domains (e.g., [49]). In addition, transposons and other repeated sequences are abundantly present between genes and in intronic sequences in human cells, generating a large amount of constitutively silent chromatin along the chromosome arms, including within genes [1,2,3,4]. To probe repair in heterochromatin, super resolution techniques have been applied to detect DSBs associated with H3K9me3-rich regions in fixed cells. For example, the Cardoso lab used super-resolution imaging techniques (3D-SIM and STED microscopy) to visualize repair in H3K9me3-rich regions in HeLa cells [49] (Figure 3). Consistent with studies in mouse cells, they detected *γ*H2AX associated with ‘silent’ histone marks at later repair steps following IR, suggesting the delayed completion of repair in heterochromatin [49]. This was confirmed by ChIP-seq analysis of *γ*H2AX in response to IR, which showed a broad enrichment of *γ*H2AX peaks in pericentromeric and other H3K9me3-enriched regions. They also estimated a reduction of DAPI intensity at later repair steps in these regions, consistent with heterochromatin de-condensation, while most silent marks were retained [49]. Similarly, single-molecule localization microscopy (SMLM) of heterochromatin and repair foci identified *γ*H2AX signals proximal to heterochromatin marks, consistent with a local relocalization of repair sites to outside of the local heterochromatic regions [69].

(+) Super resolution analyses allow the probing of heterochromatin repair responses in human cells, where heterochromatin does not form distinct domains, identifying conserved relocalization pathways. A combination of IF and FISH experiments can potentially be used to detect the behavior of specific satellites.

(−) The major caveat of the approach described here is that using H3K9me2/3 as a marker for heterochromatin does not allow discernment of different silent regions. Thus, LADs, telomeres, ribosomal sequences, silenced genes, or intragenic regions of active genes are not distinguishable from pericentromeric heterochromatin in these studies. Further, equipment and expertise required for super resolution imaging are not available in all laboratories. Finally, live imaging of multiple markers is more difficult with super-resolution techniques, limiting the dynamic information that can be extracted.

### 2.4. Chemically Induced DSBs

Not many studies have applied chemical treatments to investigate DSB repair in heterochromatin. This is likely because of the disadvantage of asynchronous damage induction and the lack of specificity coming from these approaches. A notable exception is a study in *Arabidopsis* which showed that the heterochromatin-enriched histone variant H2A.W.7 is phosphorylated by ATM to mediate the DNA damage response in heterochromatin, in response to the radiomimetic treatments Bleomycin or Zeocin [70]. Conversely, H2AX phosphorylation contributed to damage signaling mostly in euchromatin [70]). This is a unique example of differentiated damage signaling for euchromatin and heterochromatin, although H2A.W.7 and its phosphorylation are not limited to pericentromeric regions but encompass most silenced chromatin.

(+) Chemical treatments are easy to deliver as they do not require any specialized equipment.

(−) Chemical treatments induce different types of breaks [71], and they damage both euchromatin and heterochromatin. Additionally, damage is created asynchronously, complicating the study of repair dynamics.

### 2.5. DR-White Repair Cassette Inserted in Heterochromatin

Repair cassettes have been widely used to characterize repair outcomes in response to site-specific DSBs from yeast to human cells. In flies, the LaRocque lab established the direct repeat (DR)-*white* repair cassette, where HR repair of a mutated *white* gene containing an I-*Sce*I cut site (*Sce.white*), using donor sequences from a downstream truncated *white* gene (*iwhite*), results in a wild-type *white* sequence [72] (Figure 5). When I-*Sce*I is expressed in the germ line, repair products can be uniquely detected in the progeny, and HR repair leads to red-eyed flies (Figure 5). Alternatively, single-strand annealing (SSA) repair (Figure 2) results in the deletion of a gene placed between *iwhite* and *Sce.white* (e.g., a red fluorescence marker, RFP), and the loss of fluorescence in white-eyed progeny [72]. When placed in heterochromatin, the cassette is partially silenced, resulting in a mosaic phenotype [33] (variegated-red eye, Figure 3 and Figure 5). End-joining (EJ) repair outcomes (reviewed in [73,74], Figure 2) can also be characterized by sequencing across the cut site of white-eyed flies [33] (Figure 5). Finally, repair outcomes can be studied in somatic cells by PCR and sequence analyses [33].

The Karpen and LaRocque labs combined DR-*white* insertions in heterochromatin or euchromatin with genetic and live imaging experiments to establish heterochromatin repair pathways at the organismal level, including repair pathway choice in euchromatin vs. heterochromatin [33]. Using this system, they showed that NHEJ occurs with high frequency in the heterochromatin of fly tissues (which are enriched for G1 cells). Additionally, ChIP analyses of histone modifications at the repair sites and genetic approaches support the conclusion that Kdm4A demethylates H3K9me2/3 and H3K56me3 specifically at heterochromatic DSBs to promote NHEJ repair, while silencing promotes HR [76]. Importantly, Kdm4A-induced histone demethylation specifically affects heterochromatic DSBs and not euchromatic sites [76]. Finally, using a modified version of this cassette with a donor on the homologous chromosome [33,77], these studies showed that both homologous chromosomes and sister chromatids are used for HR repair in heterochromatin, albeit with a preference for the sister chromatid [33]. Notably, in fly cells, homologous chromosomes are paired in somatic cells [78], providing a readily available template for repair in addition to the sister chromatid.

(+) This system is very versatile and is applied to a genetically tractable whole organism. I-*Sce*I cleavage creates a clean and readily reparable DSB, similar to endogenous DNA breaks that arise during replication (e.g., different from complex breaks generated by exposure to radiation). Responses to breaks in euchromatin and heterochromatin can be directly compared across different strains, enabling the detection of heterochromatin-specific responses. The DR-*white* cassette can be inserted in different epigenetic contexts to study the relationship between pre-existing chromatin state and repair outcomes. Further, the use of tissue-specific promoters for I-*Sce*I or flies of different ages enables the analysis of repair outcomes across different tissues or during aging [79]. The DR-*white* system enables simultaneous study of the balance between SSA, HR, NHEJ, and alternative-EJ (Alt-EJ) in both somatic and germline cells. Additionally, the insertion of polymorphisms in the donor sequence enables the measurement of DSB repair outcomes between divergent sequences or to assess the extent of gene conversion tracts [72,80,81]. Finally, similar repair cassettes can be designed to provide a red-eyed phenotype associated with different repair pathways, as previously done for GFP in human cells [82,83,84], providing a quick readout of the pathway of interest.

(−) A potential limitation of this approach is that inserting an exogenous cassette into a genomic location might alter the behavior of that locus. For example, DR-*white* needs to be expressed in order to provide a ‘red eye’ or ‘red fluorescence’ readout, indicating that even insertions in highly silenced heterochromatin are at least partially expressed in order for the assay to work [33]. Assessing the extent to which cassette insertion affects the pre-existing chromatin state is an important step to validate the use of these constructs and provide a proper interpretation of the results. Further, DR-*white* insertions have been applied to a few genomic locations, limiting the statistical power of this approach. Satellite sequences have also not been interrogated with this system. Additionally, the impact of the DNA sequence environment on repair outcomes cannot be detected as each I-*Sce*I site is surrounded by the same sequence carried by the cassette.

Further, DR-*white* is designed to work in the whole organism, limiting the number of mutations that can be tested to investigate genetic pathways due to the relatively long time required to perform the necessary crosses. This is particularly true for live imaging experiments, where flies need to express fluorescent markers for damage (e.g., GFP-Mdc1/Mu2) and heterochromatin (e.g., mCh-HP1a), in addition to the repair cassette, the I-*Sce*I endonuclease, and the mutation of interest.

Additional challenges for the live imaging of the I-*Sce*I-induced response to DSBs inside the DR-*white* cassette relate to the higher difficulty of imaging tissues vs. cultured cells, with tissues typically requiring more cell-damaging imaging techniques and more Z-stacks, increasing the risk of photobleaching and phototoxicity [26]. For these reasons, most studies have been done in thin and transparent tissues, such as imaginal discs [33].

The asynchronous nature of I-*Sce*I-induced damage results in long imaging experiments needed for capturing relocalization events, increasing the possibility of photodamage. Further, all heterochromatic DR-*white* insertions generated so far are already positioned at the heterochromatin domain periphery at the beginning of the live imaging experiment [33]. This could be because these sites were more available for insertion of the MiMIC system used to create the DR-*white* insertions in the first place [85], or because of the difficulty of capturing early time points after I-*Sce*I induced breaks in asynchronous damage inductions. Regardless of the reason, the peripheral position of damage foci relative to the heterochromatin domain precludes the analysis of early repair dynamics.

Finally, tissues are typically characterized by significant variability in cell cycle phases, cell differentiation, and epigenetic states, all of which influence repair outcomes, and these differences cannot be easily captured at the population level. Most somatic cells are also in G0/G1, complicating the study of HR repair pathways. Indeed, cell cycle differences in whole larvae versus premeiotic germline cells might account for the different frequencies of repair outcomes observed across different studies [33].

### 2.6. Cas9-Induced DSBs in Heterochromatic Satellites

The discovery of the CRISPR/Cas9 system has sparked great progress in the field of DNA damage and repair. Primarily used for genome editing, Cas9 has also emerged as a tool of great utility to induce site-specific DSBs in euchromatin and heterochromatin, which are targeted using a single-guide RNA (sgRNA) complementary to the region of interest [86,87].

The Soutoglou lab has successfully applied Cas9 to induce DSBs in mouse chromocenters, by targeting it to the major satellite [12,88] (Figure 3). Using this system, and a combination of genetic and fixed cell imaging studies, they showed that NHEJ prevails in heterochromatin when damage is induced in G1, while HR prevails when damage is induced in S/G2 [12]. Similar to IR-induced damage in *Drosophila* cells, repair starts efficiently inside the heterochromatin domain, and this response in mouse cells is ATM-dependent [12] (Figure 4). Further, DSBs repaired by HR relocalize to the periphery of the chromocenters, forming distinct *γ*H2AX signals surrounding the DAPI-bright regions, and relocalization requires resection [12]. While resection occurs inside the domain (marked by RPA foci), Rad51 is recruited only after DSB relocalization to the periphery of the chromocenters (Figure 4). The heterochromatin domain also expands in response to Cas9-induced DSBs [12]. Finally, repair by EJ or SSA (marked by Rad52 foci), does not require relocalization [12].

Similarly, the Karpen lab used Cas9 to induce DSBs in the dodeca satellite of *Drosophila* cells [76]. They showed that Kdm4A mutation results in a higher number of foci of resection proteins (CtIP and Tosca/Exo1) inside the damaged satellite, supporting a role for Kdm4A in promoting NHEJ while inhibiting HR in these sequences [76].

(+) This is the only approach currently available to directly investigate heterochromatin repair in large satellite sequences and has the additional advantage of being available in cultured cells, facilitating live imaging, genetic approaches, and cell cycle studies. A major advantage of this system is also that a large number of breaks are induced only inside the DAPI-bright regions, so the relocalization of damage signals to outside these regions is easy to detect and quantify. Further, NHEJ proteins that do not form foci in response to IR are detectable in response to Cas9-induced breaks, given that the large amount of damage in the satellite amplifies the signal. Other repeated sequences can be targeted with similar approaches, including centromeres [12,89] or rDNA [90,91,92], enabling the exploration of several domains characterized by repeated DNAs.

(−) Similar to other endonuclease-based systems, Cas9-based systems induce damage asynchronously. Thus, they are less suitable for focus kinetic and dynamic analyses relative to IR-induced breaks. Breaks are only induced in satellite DNAs, so direct comparisons with responses in euchromatin have not been done with this system. Error-free repair results in re-cleavage of the damage by Cas9, potentially affecting the frequency at which each repair pathway is utilized. Using Cas9 to induce damage might also affect repair kinetics and outcomes. For example, microhomology-mediated end joining (MMEJ) pathways seem to require unusually long kinetics when damage is induced with Cas9 [93]. Similarly, the Fanconi Anemia pathway is involved in repairing Cas9-induced damage, while this pathway is typically required only in the presence of inter-strand crosslinks or stalled replication forks [94]. In yeast, repair of Cas9-induced breaks at the boundary between single-copy and CTG repeats results in more extensive resection and a higher frequency of SSA repair on the side containing repeated sequences, leading to unexpectedly large chromosomal deletions of the CTG repeat, and suggesting specific effects of Cas9 on repair in repeated sequences [95].

### 2.7. Biochemical Fractionation of Heterochromatin

Biochemical approaches have been broadly applied to the study of heterochromatin repair, for example for identifying components that co-immunoprecipitate with heterochromatin histone marks or associated proteins in response to DNA damage [10,96,97]. Some of the most appealing techniques use fractionation approaches to separate heterochromatin from euchromatin and isolate heterochromatin-specific repair components. For example, the Jeggo and Goodarzi labs took advantage of the resistance of heterochromatin to micrococcal nuclease (MNase) digestion to isolate chromatin fractions containing silent histone marks and associated proteins [9,53]. They also discovered that damaged heterochromatin is more susceptible to MNase digestion than undamaged heterochromatin, reflecting a relaxation of the domain [53]. Similar fractionation studies have identified a role for Rad6 in HP1α degradation to promote heterochromatin relaxation and HR repair [98]. Biochemical approaches have also established that phosphorylated HP1β is released from damaged chromatin, promoting the recruitment of the histone acetyltransferase Tip60 and ATM activation [97].

Of note, MNase-resistant fractions are not only enriched for pericentromeric heterochromatin, but also for other silenced regions of the genome (e.g., LADs and silenced genes). Thus, the use of chromatin fractionation with MNase digestion does not enable a direct characterization of repair in pericentromeric sequences. Nevertheless, this technique provided important complementary approaches to live and fixed cell imaging studies for the characterization of repair components acting in pericentromeric regions.

(+) Chromatin fractionation and other biochemical approaches have been successfully applied to identify components that associate with heterochromatin in response to damage, heterochromatin relaxation, and the components mediating this response.

(−) Nuclease-resistant fractions are largely free of euchromatic contamination in the absence of DNA damage, but fractions that are slightly more sensitive to MNase are typically enriched for both euchromatic and heterochromatic histone marks [53]. Additionally, even the most MNase-resistant fractions contain different silenced sequences, not just pericentromeric heterochromatin. Thus, MNase-dependent fractionation methods are not robust enough to clearly separate euchromatin and heterochromatin biochemically, and thus to distinguish between repair components uniquely associated with each of these domains.

## 3. Perspectives

The variety of techniques discussed above have driven the heterochromatin repair research in the past decade, leading to exciting new discoveries (Table 1). Live and fixed cell imaging after irradiation, the use of repair cassettes inserted into heterochromatin, and Cas9-induced DSBs, have enabled the identification of heterochromatin-specific responses. Biochemical approaches have provided important complementary tools for these discoveries. Here, we reflect on some of the limitations of these techniques and suggest possible areas of development to further deepen our understanding of this field in future studies.

### 3.1. Live and Fixed Cell Imaging

Cell imaging techniques are rapidly evolving, with advances in instrumentation and new tools for biomolecule visualization. For example, the application of extrinsically fluorescent protein tags (i.e., SNAP [99], Halo [100], and CLIP [101] (reviewed in [102]) provides signals that are typically brighter and more photostable than those detected with intrinsically fluorescent tags (e.g., GFP, mCherry). Applying extrinsically fluorescent tags to the study of heterochromatin and repair proteins would enable faster and longer kinetics, with a more extensive exploration of heterochromatin dynamics. Collecting more time points during focus relocalization would also enable the tracking of cells for multiple hours after IR, and, for example, establish what happens to the repair site position or heterochromatin decondensation once repair is complete. More frequent imaging would also facilitate the automated tracking of repair foci by reducing the frequency of ambiguous tracks [51,52], and enable the study of local chromatin movements [30]. Additionally, small X-ray sources can be directly mounted on a microscope to enable the detection of very early repair responses [103,104].

In fixed cell imaging, a proximity ligation assay (PLA) can be applied to detect low abundance proteins or specific post-translational modifications at heterochromatic repair sites [105].

Further, both live and fixed cell imaging studies would greatly benefit from the development of fully automated methods to detect repair focus positions relative to the heterochromatin domain, dramatically reducing the processing time for image/movie analyses.

In addition, several studies have identified phase separation properties associated with the heterochromatin domain and repair foci [106,107,108,109,110,111], but the impact of these properties on heterochromatin repair is unknown. Future studies can combine live imaging approaches with fluorescent recovery after photobleaching (FRAP) or optogenetic tools that induce phase separation (reviewed in [112,113]) to better characterize the phase separation properties of the heterochromatin domain and repair foci, and the role(s) of these properties in repair.

Finally, non-coding RNAs (ncRNAs) have been associated with both heterochromatin establishment [114] and DSB responses (reviewed in [115,116,117]), but their roles in the DSB repair of pericentromeric regions remains unexplored. The application of recently-developed tools to investigate the role of ncRNAs in repair, such as antisense oligonucleotides (ASOs) [118,119,120] coupled with site-specific DSB systems described in this review, provides new opportunities to explore these responses in heterochromatin.

### 3.2. Site-Specific DSB Induction Systems

The use of restriction enzymes that directly cut the genome in different positions revolutionized our understanding of repair responses with genome-wide approaches. For example, in the DIvA (DSB inducible via *Asi*SI) cell system developed by the Legube lab [121], the restriction endonuclease *Asi*SI cuts the genome at more than a hundred sites. Site-specific DSBs have been used for studying: DSB resection, using quantitative real time PCR or END-seq [122,123,124]; repair pathway choice, using ChIP-seq analysis of HR and NHEJ components [38,40,125]; and the spreading of *γ*H2AX signaling by loop extrusion, using Hi-C approaches [126]. However, the DIvA system has not been applied to pericentromeric heterochromatin, because DSBs are not efficiently induced by *Asi*SI in these sequences due to DNA methylation and chromatin compaction [121,123]. An important goal for future studies is to extend site-specific DSB systems to pericentromeric sequences, such as by working in model systems that lack DNA or using approaches that overcome the barrier to digestion.

### 3.3. Repair Cassettes

In addition to generating new repair cassettes for *Drosophila* repair studies (such as to explore canonical end-joining and Alt-EJ pathways with ‘red-eyed’ readouts), it would be important to develop repair cassettes that do not rely on transcription for a readout.

Applying these tools to multiple genomic sites simultaneously would also greatly improve the statistical power of these approaches. For example, in a recent study, the van Steensel lab integrated a multiplexed reporter system in thousands of genomic loci with Cas9, and explored the relative contribution of three DSB repair pathways (NHEJ, MMEJ, or single strand template repair (SSTR)), in different chromatin contexts, including in silenced chromatin [37]. These studies identified a preferential usage of MMEJ in facultative heterochromatin (marked by H3K27me3), although pericentromeric regions were not directly investigated [37]. Extending similar tools to pericentromeric heterochromatin would provide new opportunities for discovery.

### 3.4. Exploring the Heterochromatin Damage Proteome with Biochemical Approaches

The isolation of damaged heterochromatin with biochemical approaches would enable the identification of new components required for heterochromatin repair. Chromatin fractionation techniques have greatly helped dissect heterochromatic specific regulators and study chromatin decompaction. However, the current methods do not achieve a ‘clean’ separation between euchromatin and heterochromatin. As an alternative approach, immunoprecipitation techniques targeting HP1 have been successfully applied to isolate heterochromatin proteins in different model systems [127,128,129,130,131,132]. Similarly, proximity-dependent biotin identification (BioID) techniques [133], have been applied to identify components associated with silenced chromatin domains, including LADs [133,134], telomeres [135,136], and H3K9me3-enriched-regions [137]. While these approaches did not (or not exclusively) address repair in pericentromeric heterochromatin, applying similar purifications in response to DNA damage can help reveal the heterochromatin-specific proteome for repair. An alternative chromatin fractionation method to MNase digestion took advantage of the sensitivity of different types of chromatins to sonication, and successfully separated H3K9me3 from H3K27me3-enriched fractions for proteomic and sequencing analysis [138].

Additionally, purification techniques can be theoretically applied to Cas9-induced damage specifically in euchromatin or heterochromatin satellites, isolating the components that uniquely contribute to repair of pericentromeric regions. Together, biochemical fractionation and immunoprecipitation approaches are attractive options to isolate novel components participating in heterochromatin repair and can be combined with complementary live and fixed cell imaging studies described in this review, and further dissect the contribution of different components to repair in pericentromeric regions.

## 4. Conclusions

The ability to induce DSBs in heterochromatin with different approaches, and to detect repair with a variety of techniques, has significantly enhanced our understanding of heterochromatin repair mechanisms in the past decade. These studies mostly relied on; (i) live and fixed imaging after irradiation in systems with distinct heterochromatin domains; (ii) the use of reporter cassettes inserted in heterochromatin; (iii) the induction of DSBs in satellite sequences with Cas9; and (iv) biochemical approaches to enrich for heterochromatin fractions (Figure 3 and Table 1). While these techniques enabled major advances in our understanding of heterochromatin repair, many open questions remain. Improving these approaches, along with adapting the tools recently developed for euchromatin repair studies to heterochromatin domains, will enable a deeper understanding of heterochromatin repair mechanisms. Urgent unanswered questions include understanding chromatin responses in heterochromatin by next-generation sequencing, establishing repair outcomes across multiple heterochromatic sites, identifying the role of phase separation and ncRNAs in heterochromatin repair, and advancing our understanding of the spatial dynamics of heterochromatin repair in human cells. With this ever-growing toolbox, we expect to see exciting progress in our understanding of heterochromatin repair mechanisms across different model systems, damage sources, epigenetic states, and sequence composition, in the coming years.

## Figures and Tables

**Figure 2 genes-13-00529-f002:**
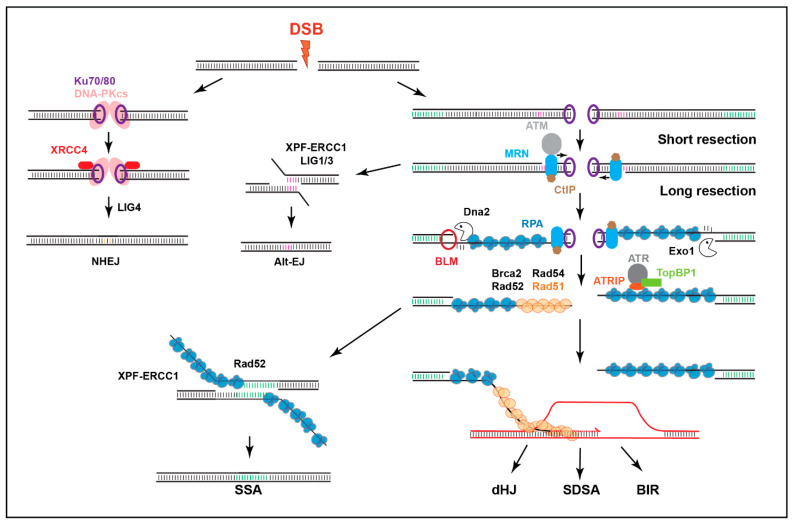
Main DSB repair mechanisms. DSBs are mainly repaired by HR (mostly in S/G2) or NHEJ. NHEJ involves no or minimal processing of the DSB. HR relies on DSB resection to form ssDNA that ‘invades’ homologous sequences on the sister chromatid or homologous chromosomes to enable DNA synthesis and repair. In mitotic cells, synthesis-dependent strand annealing (SDSA) is the most common HR pathway, although break-induced replication (BIR) also frequently occurs near telomeres. Double-Holliday junction (dHJ) repair is better characterized in meiosis. Extensively resected DSBs can also utilize the Rad51-independent pathway of single strand annealing (SSA) when repeated sequences are available, while alternative-end joining pathways (Alt-EJ), such as MMEJ, involve limited resection to expose micro-homologies.

**Figure 5 genes-13-00529-f005:**
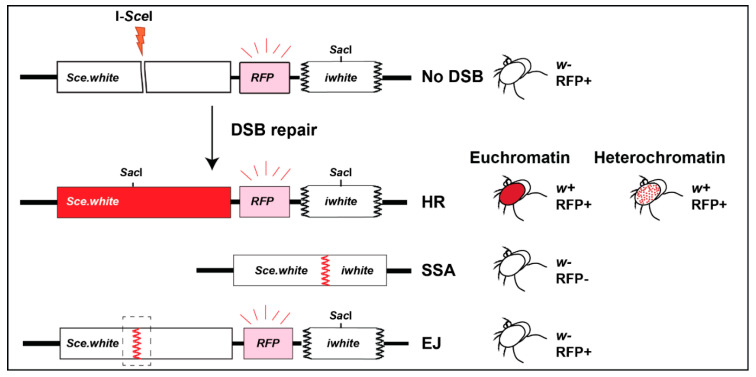
Schematic representation of DR-white cassette and repair outcomes. The DR-*white* reporter is composed of an upstream nonfunctional white gene (*Sce.white*) containing an I-*Sce*I target sequence, a red fluorescent marker (RFP), and a downstream truncated nonfunctional white gene (*iwhite*) (Adapted from [75]). *Sce.white* contains a premature stop codon as part of the I-*Sce*I recognition sequence. Repair of DSBs induced by I-*Sce*I in the germline of male flies can be detected in the progeny. HR repair of euchromatic insertions results in red eyes, and HR of heterochromatic insertions results in variegated-red eyes [33]. SSA results in *white-*, RFP- flies. Uncut, NHEJ, or Alt-EJ result in *white-*, RFP+ progeny, and different repair products can be distinguished by sequencing across the cut site.

**Table 1 genes-13-00529-t001:** Summary of techniques available to study heterochromatin repair, including main advantages and disadvantages.

Technique Applied to Heterochromatin	Advantages	Disadvantages	Refs.
**Imaging of IR-induced foci**	Nearly synchronous DSB induction facilitates the study of repair kinetics and 3D dynamics Easy detection of heterochromatin ‘domains’ in *Drosophila* and mouse cells Responses in euchromatin and heterochromatin can be directly compared within the same cells Easy genetic and biochemical approaches in cultured cells	Not easily applicable to human cells that lack distinct heterochromatin domains DNA lesions other than DSBs are induced Not sequence-specific, and low resolution Live imaging conditions need to be optimized to limit photodamage and photobleaching Some repair proteins do not form visible foci Limited components can be analyzed simultaneously, particularly in live imaging experiments Tedious quantifications Specific equipment required (X-ray or ɣ-ray irradiator)	[9,10,15,25,26,28,50,51,52,53,54,55,56,57]
**Imaging after laser or heavy-ion irradiation**	Easy quantifications Responses in euchromatin and heterochromatin can be directly compared within the same cells Low abundance proteins easily detectable at damage sites Data can be collected right after damage induction	Can induce protein damage, affecting relocalization pathways Multiple types of DNA lesions and clustered DSBs are induced Pretreatments for UV laser experiments can alter chromatin responses Non-physiological damage distribution Not sequence-specific, and low resolution Not easily applicable to small nuclei Specific equipment required (laser mounted on microscope or access to a synchrotron)	[10,32,46]
**Super-resolution imaging**	Addresses the behavior of repair foci relative to ‘silent’ chromatin marks at high resolution, including in cell types that lack heterochromatin ‘domains’	The use of H3K9me2/3 as a marker for heterochromatin does not enable distinguishing between different silent regions Live imaging is difficult Specific equipment (e.g., 3D-SIM, STED microscope) and expertise required	[49,69]
**Chemically induced DSBs**	Easy to administer Does not require specialized equipment	Asynchronous damage induction and lack of specificity Creation of different types of breaks	[70]
**DR-*white* repair cassette**	Enables studying site-specific DSB repair outcomes and repair pathways Responses in euchromatin and heterochromatin can be compared across different strains. Applicable to a genetically tractable whole organism I-*Sce*I-induced cleavage mimics endogenous DSBs Tissue specific promoters for I-*Sce*I enable studying specific tissues and ages Enables the study of repair outcomes in both somatic and germline cells	The insertion of a transcribing exogenous cassette can affect the behavior of the heterochromatic locus A limited number of sites have been tested Genetic experiments are time consuming when several markers need to be tracked Live imaging of thick tissues is challenging Asynchronous damage induction complicates the study of kinetic and dynamic responses Cell cycle differences across tissues can affect repair outcomes	[33,76]
**Cas9-induced DSBs in satellites.**	Only system to directly damage satellite DNA Easy quantifications Available in cultured cells, facilitating live imaging, genetic approaches and cell cycle studies Low abundance proteins easily detectable at damage sites	Asynchronous damage induction complicates the study of kinetic and dynamic responses Repair pathway usage can be affected by re-cleavage and by the use of Cas9 Does not enable direct comparison with responses in euchromatin	[12,76,88]
**Biochemical fractionation**	Enables the study of components mostly associated with ‘silent’ chromatin	Does not uniquely identify pericentromeric heterochromatin	[9,53,96,98]

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
