# Peer review of "An Expanding Toolkit for Heterochromatin Repair Studies"

_genes, 2022, doi:10.3390/genes13030529_

Round 1

Reviewer 1 Report

Well written and interesting manuscript. Accept as it is.

Author Response

Well written and interesting manuscript. Accept as it is.

Response: We really appreciate the positive feedback!

Reviewer 2 Report

Rawal et al.’s review article is comprehensive and features most state-of-the art techniques to monitor heterochromatin repair dynamics. The text is well written, and overall the figures are clear and properly depict the different tools available to study heterochromatin.

This review would however benefit from expanding into the field of antisense oligonucleotides (ASOs) as tools to inhibit transcripts that are known to regulate DNA damage signaling at telomeres and other heterochromatin sites. Targeting TERRA and telomeric dilncRNAs at these heterochromatic regions have been shown to modulate the kinetics and efficiency of repair (doi: 10.1016/j.cell.2017.06.017 and doi: 10.1038/ncomms13980); and more recently these telomeric ASO tools have been successfully utilized as proof of concept to targeting telomere dysfunction pathology (doi: 10.1038/s41467-019-13018-3). This ASO technology may sit well among the tools so far described in Figure 2.

As for the use of BioID (or proximity-dependent biotin identification, originally developed by Roux et al. 2012), it is worth adding that this valuable tool has been used to identify LADs, where for instance mutations in Lamin A directly impacts LAP2 alpha, ultimately destabilizing heterochromatic regions where telomere DNA damage is induced (example, doi: 10.7554/eLife.07759).

Author Response

Rawal et al.’s review article is comprehensive and features most state-of-the art techniques to monitor heterochromatin repair dynamics. The text is well written, and overall the figures are clear and properly depict the different tools available to study heterochromatin.

This review would however benefit from expanding into the field of antisense oligonucleotides (ASOs) as tools to inhibit transcripts that are known to regulate DNA damage signaling at telomeres and other heterochromatin sites. Targeting TERRA and telomeric dilncRNAs at these heterochromatic regions have been shown to modulate the kinetics and efficiency of repair (doi: 10.1016/j.cell.2017.06.017 and doi: 10.1038/ncomms13980); and more recently these telomeric ASO tools have been successfully utilized as proof of concept to targeting telomere dysfunction pathology (doi: 10.1038/s41467-019-13018-3). This ASO technology may sit well among the tools so far described in Figure 2.

Response: We appreciate this recommendation. Indeed, ASOs can be used to explore the effects of heterochromatin transcription in repair, which is still unexplored in pericentromeric sequences. Thus, we have now included a reference to these techniques as part of the ‘Perspectives’ section (lines 556-562).

As for the use of BioID (or proximity-dependent biotin identification, originally developed by Roux et al. 2012), it is worth adding that this valuable tool has been used to identify LADs, where for instance mutations in Lamin A directly impacts LAP2 alpha, ultimately destabilizing heterochromatic regions where telomere DNA damage is induced (example, doi: 10.7554/eLife.07759).

Response: This is another excellent point. We have now included a description of BioID techniques applied to different silenced domains in the ‘Perspecives’ section, including highlighting the importance of extending these studies to pericentromeric sequences (lines 597-600).

Reviewer 3 Report

In the manuscript entitled “An expanding toolkit for heterochromatin repair studies” the authors review the tools, approaches, and methods currently available to investigate DSB repair in pericentromeric heterochromatin-enriched regions. Whatever is described in the manuscript is of practical importance towards understanding the mechanisms that maintain genome integrity. However, the manuscript needs to be restructured. To better “digest” the content of the manuscript, description of methods with their advantages and disadvantages in Sections 1–7 can be presented in a tabular format.

In Fig. 2A legend the phrase “treated with 5 Gy X-Ray” is recommended to write as “treated with the dose of 5 Gy of X-rays”. What is the dose and the type of IR was used in the experiment depicted in Fig. 2B? Also, the magnified images in this figure should be supplemented with the time points after irradiations (10 and 60 min respectively; is this correct?). The same situation with Fig. 2C – some more information about irradiation conditions is needed. Perhaps this figure represents “fixed cell imaging” technique as well. Again, the phrase “after 10 Gy of X-ray” (Fig. 2D legend) can be written as “after X-ray exposure with the dose of 10 Gy”. The purpose of experiments depicted in Figs. 2E, G and F is unknown and should be concisely explained in the legend.

In Section 1, Line 99 – “?H2Av” is likely to be “?H2AX” (see also lines 207, 208, 231, 277, 278, 282, etc.).

Lines 96-100 – which cells were studied for DSB detection, signaling, resection, Rad51 loading and strand invasion? If Drosophila and mouse cells only – cell types should be specified.

Line 198 – “X-ray irradiator lamp or Cesium irradiator” replace with “X-ray machine or 137Cs-irradiator”.

Abbreviations like IR, ATRIP, HP1a, 3D-SIM, DR … as well as the terms like Cas9, gRNA, MNase … may not be well known for the reader who is less familiar with this topic and they need to be concisely specified ether in text or in Appendix section.

Author Response

In the manuscript entitled “An expanding toolkit for heterochromatin repair studies” the authors review the tools, approaches, and methods currently available to investigate DSB repair in pericentromeric heterochromatin-enriched regions. Whatever is described in the manuscript is of practical importance towards understanding the mechanisms that maintain genome integrity. However, the manuscript needs to be restructured. To better “digest” the content of the manuscript, description of methods with their advantages and disadvantages in Sections 1–7 can be presented in a tabular format.

Response: We agree that a table would make this section easier to ‘digest’. In the revised manuscript, we have added a summary of the main advantages and disadvantages of different techniques as a new Table 1, page 15, lines 608-610).

In Fig. 2A legend the phrase “treated with 5 Gy X-Ray” is recommended to write as “treated with the dose of 5 Gy of X-rays”. What is the dose, and the type of IR was used in the experiment depicted in Fig. 2B? Also, the magnified images in this figure should be supplemented with the time points after irradiations (10 and 60 min respectively; is this correct?). The same situation with Fig. 2C – some more information about irradiation conditions is needed. Perhaps this figure represents “fixed cell imaging” technique as well. Again, the phrase “after 10 Gy of X-ray” (Fig. 2D legend) can be written as “after X-ray exposure with the dose of 10 Gy”. The purpose of experiments depicted in Figs. 2E, G and F is unknown and should be concisely explained in the legend.

Response: The figure legend (now Fig. 3, lines 122-133) and the figure itself have now been edited to include the type of IR, doses and time points, as suggested. The phrasing related to dose treatment has also been edited as suggested. Further, we have expanded the description of the legend to improve clarity.

In Section 1, Line 99 – “?H2Av” is likely to be “?H2AX” (see also lines 207, 208, 231, 277, 278, 282, etc.).

Response: Thank you for noticing these typos. We fixed all of them in the new version of the manuscript (now lines 173, 233,235,251,305-312,451, and 570).

Lines 96-100 – which cells were studied for DSB detection, signaling, resection, Rad51 loading and strand invasion? If Drosophila and mouse cells only – cell types should be specified.

Response: Thank you for pointing out that this section was not clear. We have now better introduced the model system in which distinct markers for repair progression are used between Drosophila and mammalian cells (lines 85-86, 95-96), when these markers are different. We have also now further clarified the model system in which each experiment has been done in the subsequent sections (lines 103, 111-112), to improve clarity.

Line 198 – “X-ray irradiator lamp or Cesium irradiator” replace with “X-ray machine or 137Cs-irradiator”.

Response: We have revised the text accordingly (line 223).

Abbreviations like IR, ATRIP, HP1a, 3D-SIM, DR … as well as the terms like Cas9, gRNA, MNase … may not be well known for the reader who is less familiar with this topic and they need to be concisely specified ether in text or in Appendix section.

Response: Thank you for this excellent point. We have now added a glossary with details better explaining different proteins and techniques (lines 636-637). Additional explanations have been added to the main text (e.g., line 347). We also note that we have now included a new Figure 2 with the main DSB repair pathways and some of the most important components involved, which further facilitates the understanding of the function of repair proteins mentioned in the text for readers that are not familiar with the topic. We agree that these changes greatly improve the manuscript.

Reviewer 4 Report

In the review by Rawal et al., the different tools currently available to study the repair mechanisms of DSB in heterochromatin are described in detail. The article focuses mainly on studies in Drosophila and mammalian cells. A description of the heterochromatic regions in the chromosomes is presented, the mechanisms that the cell has to respond to this type of insult are described in a general way, also how this DNA repair occurs and the different experimental ways of causing DSB in the cells, all focused on heterochromatin.

In a precise and simple way, they describe the techniques that exist to study DSB in heterochromatin. I think this is a good review and that it can be used as a reference for groups that carry out studies on the mechanisms of response and repair of DNA damage, in  general.

The review meets the merits to be published in Genes, however I have some suggestions that I think can make the article more accessible to an audience not so specialized in heterochromatin and DNA repair.

My points are the following:

1.-Although the figures presented in the article are adequate, I think that a figure is needed showing the NHEJ and HR mechanisms, indicating the factors that participate in it.

2.- During the text, a large number of cellular factors and components that participate in the DDR and in damage repair are briefly mentioned, as well as different components of heterochromatin and enzymes whose substrate are nucleosomes and other components of heterochromatin. I suggest to add a glossary on all these players in a part of the article, in similar way as glossary  is reported in reviews in other journals It would be very useful for the reader.

3.-One point that is not described in the paper, at least not clearly, is that if the DSB repair mechanisms are the same in heterochromatin and euchromatin. Are the techniques to study DSB in euchromatin and heterochromatin the same or are there some that are better in one case than another? This is a point that this paper should discuss.

Author Response

In the review by Rawal et al., the different tools currently available to study the repair mechanisms of DSB in heterochromatin are described in detail. The article focuses mainly on studies in Drosophila and mammalian cells. A description of the heterochromatic regions in the chromosomes is presented, the mechanisms that the cell has to respond to this type of insult are described in a general way, also how this DNA repair occurs and the different experimental ways of causing DSB in the cells, all focused on heterochromatin.

In a precise and simple way, they describe the techniques that exist to study DSB in heterochromatin. I think this is a good review and that it can be used as a reference for groups that carry out studies on the mechanisms of response and repair of DNA damage, in general.

The review meets the merits to be published in Genes, however I have some suggestions that I think can make the article more accessible to an audience not so specialized in heterochromatin and DNA repair.

My points are the following:

1.-Although the figures presented in the article are adequate, I think that a figure is needed showing the NHEJ and HR mechanisms, indicating the factors that participate in it.

Response: We thank the reviewer for this suggestion. The revised version contains new Figure 2 with the main DSB repair pathways and proteins, as suggested (line 87-94).

2.- During the text, a large number of cellular factors and components that participate in the DDR and in damage repair are briefly mentioned, as well as different components of heterochromatin and enzymes whose substrate are nucleosomes and other components of heterochromatin. I suggest to add a glossary on all these players in a part of the article, in similar way as glossary is reported in reviews in other journals It would be very useful for the reader.

Response: We have now included glossary for proteins participating in DDR, as also requested by reviewer 3. In the interest of keeping this section concise, we focused on components that were only briefly mentioned in the text. Those that are not in the glossary, are now better introduced in the main text (see for example lines  84-86 and 134). We agree that these changes, in addition to the new Figures 2 and 4 should greatly help the readers to follow different mechanisms and components mentioned in the text. Thank you for the useful suggestion!

3.-One point that is not described in the paper, at least not clearly, is that if the DSB repair mechanisms are the same in heterochromatin and euchromatin. Are the techniques to study DSB in euchromatin and heterochromatin the same or are there some that are better in one case than another? This is a point that this paper should discuss.

Response: We are particularly grateful to this reviewer for bringing up this important point. Indeed, some of the approaches described in this review are suitable for direct comparisons between repair in euchromatin and heterochromatin (live imaging of IR responses, laser/heavy particles-induced damage, repair cassettes), enabling the identification of heterochromatin-specific effects. Applying these approaches established, for example, the heterochromatin-specific or heterochromatin-predominant role of relocalization pathways, pKap1, and Kdm4A in DSB repair. Other techniques, such as the use of Cas9 directed towards heterochromatic satellites, do not enable direct comparisons with euchromatin. We now bring up these points across the review, including highlighting them as part of the new table that includes strengths and weaknesses of different techniques (see lines 147-150, 176, 192-198, 268, 375, 385-86, 474-75 and table 1 at line 608). We agree that including these considerations greatly improve the review.

Round 2

Reviewer 2 Report

The authors have significantly improved the manuscript. 

I recommend publication. 

Reviewer 3 Report

Significant improvement.